

# Governing Change: A Dynamical Systems Approach to Understanding the Stability of Environmental Governance

Nusrat Molla[1], John DeIonno[2], Thilo Gross[3], and Jonathan Herman[2]

[1]Land, Air, and Water Resources, University of California, Davis
[2]Civil and Environmental Engineering, University of California, Davis
[3]Helmholtz Institute for Functional Marine Biology, University of Oldenburg

**Correspondence:** Nusrat Molla (njmolla@ucdavis.edu)

**Abstract.** The ability to adapt to social and environmental change is an increasingly critical feature of environmental governance. Yet, an understanding of how specific features of governance systems influence how they respond to change is still limited. Here we focus on how system features like diversity, heterogeneity and connectedness impact stability, which indicates the system's capacity to recover from perturbations. Through a framework that combines agent-based modeling with 5 "generalized" dynamical systems modeling, we model the stability of thousands of governance structures consisting of groups of resource users and non-government organizations interacting strategically with the decision centers that mediate their access to a shared resource. Stabilizing factors include greater effort dedicated to venue shopping, and a greater fraction of non-government organizations in the system. Destabilizing factors include greater heterogeneity among actors, a greater diversity of decision centers, and greater interdependence between actors. The results suggest that while complexity tends to be 10 destabilizing, there are mitigating factors that may help balance adaptivity and stability in complex governance. This study demonstrates the potential in applying the insights of complex systems theory to managing complex and highly uncertain human-natural systems in the face of rapid social and environmental change.

## 1 Introduction

Social-ecological outcomes such as sustainability, resilience, and equity, are ultimately the product of a complex set of in-
teractions among networks of autonomous actors self-organizing to address interconnected issues – or in short, governance
(Carlisle and Gruby, 2019; Klijn and Snellen, 2009; Koppenjan and Klijn, 2004; Pahl-Wostl, 2009; Stephan et al., 2019). A
diverse literature has emerged to explore governance as a complex system, which breaks with the traditional notion of gover-
nance as a linear and centrally managed process of planning and execution (van Buuren et al., 2012; Klijn and Snellen, 2009;
Pahl-Wostl, 2009). Instead, complex governance emphasizes interactions among mutually dependent actors, a structure that is
at least partially self-organized rather than externally imposed (Pahl-Wostl, 2009; Lubell and Morrison, 2021; Stephan et al.,
2019), and cross-scale feedbacks. Evolution in the structure and function of governance is understood to be the norm rather than
the exception (Thiel et al., 2019). This conceptualization of governance has been explored from various perspectives, including
adaptive governance (Folke et al., 2005), collaborative governance (Ansell and Gash, 2008; Ansell, 2012; Gerlak et al., 2012),





multi-level governance (Bache et al., 2016; Hooghe and Marks, 2002; Liesbet and Gary, 2003; Newig and Fritsch, 2009), and
polycentric governance (Ostrom et al., 1961; Ostrom, 2010; McGinnis and Ostrom, 2012; Carlisle and Gruby, 2019).

A central question regarding complex governance is how its structure impacts its function. For example, multiple autonomous but interdependent decision centers, a defining feature of polycentric governance (Ostrom et al., 1961; Ostrom, 2010; McGinnis and Ostrom, 2012), has been ascribed numerous benefits, such as effective production and provision of diverse public goods (Ostrom et al., 1961; Pahl-Wostl, 2009), and greater ability to adapt to a changing environment (Folke et al., 2005;
Bixler et al., 2016; Pahl-Wostl, 2009; da Silveira and Richards, 2013). In Ostrom's institutional design principles, multi-level, nested governance is associated with robust institutions for maintaining the commons (Ostrom, 1990). A greater diversity of stakeholders is thought to yield better environmental outcomes (Newig and Fritsch, 2009) and more flexible and responsive governance processes that are better able to navigate external complexity and change (Craig et al., 2017; Koppenjan and Klijn, 2004). However, much of the focus has been on associating system outcomes with collaborative or polycentric governance
as a whole rather than with specific factors, such as diversity in institutions and decision centers, heterogeneity among stakeholders, or connectivity among policy actors. This is perhaps because case studies make it challenging to independently test the effect of these different features. Understanding how these features relate to different governance outcomes with greater specificity is important in diagnosing cases in which the expected benefits associated with complex governance do not materialize (Carlisle and Gruby, 2019; McGinnis and Ostrom, 2012). This study disentangles the effect of these different features
of complex governance systems by developing a modeling approach that allows for generating and analyzing the system-level outcomes associated with ensembles of resource governance systems with different configurations.

A system-level outcome of particular interest is how governance systems respond to change, given that constant change is a central feature of complex systems. This study will focus on this question with respect to the concept of local asymptotic stability of dynamical systems, hereinafter referred to as stability. Stability refers to the ability of the system to retain its structure and
function in the face of perturbations in the variables controlled by the governance system (Guckenheimer and Holmes, 1983). In addition to being well-defined mathematically, stability is considered an important feature of governance systems, if not a universally desirable one. On the one hand, stability in governance arrangements allows people and organizations to learn about one another, experiment, and make long-term investments (Craig et al., 2017; Pahl-Wostl, 2009). It allows for the accumulation of wellbeing and resources when external change is slow and predictable, and reduces transaction costs and increases returns
from cooperation (North, 1990, 2005). On the other hand, stability can correspond to rigidity, in which governance systems fail to respond to internal changes (Carpenter and Brock, 2008; Craig et al., 2017). Stability also serves as a prerequisite for ecological resilience, which emphasizes the ability of a system to absorb perturbations without changing structurally (Holling, 1973), but may conflict with adaptive capacity, which emphasizes transformation (Folke et al., 2002; Carpenter and Brock, 2008). Understanding stabilizing factors in resource governance systems therefore also gives insight into their resilience and
adaptive capacity.

While the question of how features of governance correspond to stability has not yet been addressed with much specificity or precision, the factors that lead to stability in complex systems has long been debated in the complexity literature, particularly in the context of ecosystems. For example, increased complexity in food webs, in terms of species diversity and their connectivity,



has been shown to lead to decreased robustness (May, 1972; May et al., 2008), while certain predator-prey ratios have been
found to be stabilizing (Bambach et al., 2002). Therefore, in addition to a better understanding of complex governance, this
study provides insight into whether the principles for stability that have been discovered in other complex systems generalize
to social-ecological systems.

## 2   Modeling Approach

This study focuses on common-pool resource governance with a resource that is subtractible, such as a groundwater aquifer,
though the model could be parameterized for public goods instead. The overall modeling approach consists of defining the
structure of the dynamical system in terms the different components of the governance system and their interactions, using
generalized modeling to analyze stability without specifying functional forms. This method is particularly suited to gaining
general insights about a system despite the large uncertainties that may exist, particularly in social systems, by allowing for
studying an ensemble containing several thousand realizations of variants of the system structure. Computing the stability of
the diverse system realizations in the ensemble thus allows us to identify underlying principles for stability.

The modeling framework consists of state variables representing the state of the shared resource and the organizational
capacity of three types of entities: 1) resource user organizations or interest groups, which directly impact or are impacted
by the resource state, and can represent both extractive and non-extractive users, 2) non-government organizations, which do
not directly impact and are not directly impacted by the resource state but still have interests in the system (e.g. nonprofits,
advocacy and education groups not directly tied to a particular resource use), and 3) decision centers, which have the ability
to directly mediate resource users' interactions with the resource. Organizational capacity refers to resources like volunteer
or staff labor, access to legal, technical, or administrative expertise, funds, or grassroots engagment. Resource users and non-
government organizations will be referred to collectively as "actors" since they have an inherent stake in the system and are
being modeled as strategic and self-organizing agents.
This model focuses on the processes that take place among the stakeholders themselves, such as collaboration and undermin-
ing, resistance and support, and lobbying (Table 1). This bottom-up perspective is chosen because of the under-representation
of actors' agency in making and influencing decisions and pursuing their goals in the polycentric governance literature, which
tends to focus solely on structure and exclude entities that lack the authority to create policies, though this is changing with
concepts like institutional navigation (Lubell and Morrison, 2021) and commoning (Dobbin, 2021; Villamayor-Tomas and
García-López, 2018). Modeling actors' ability to influence the effectiveness of policies or the capacity of other actors or
decision centers to fulfill their missions captures the informal arenas and resistance of various sorts that can be pivotal in
determining outcomes in resource governance, especially where state capacity and coherence is lacking (McCarthy, 2002).
This perspective is therefore suited to a complex systems approach since it is these often informal interactions that ultimately
manifest as coalitions and movements to drive change in governance systems. The next few sections will outline the processes
that are modeled for each type of state variable.



| Process | Target | Drivers | Example Parameters |
|---|---|---|---|
| Resource Natural Gain/Loss | Resource state | Resource state | Sensitivity of natural gain to current resource state |
| Recruitment or External Support/Attrition | Actor or Decision Center capacity | Actor or Decision Center capacity | Sensitivity of capacity gain to current capacity |
| Extraction/Access | Resource | Policies supporting or reducing extraction, Actors' policy support/resistance | Share of extraction by each user, sensitivity of extraction/access to current state |
| Policies supporting or reducing extraction | Extraction | Decision center capacity, Actors' policy support/resistance | Sensitivity of extraction to policy |
| Actor Policy Support/Resistance | Policies supporting or reducing extraction | Actor capacity, Actor strategy | Sensitivity of policy effectiveness to Actor efforts to resist/support it |
| Collaboration/ Undermining | Actor capacity | Actor capacity, Actor strategy | Share of actor capacity gain from collaborating with others, sensitivity of actor capacity to efforts to support them |
| Decision Center Support/Resistance | Decision Center capacity | Decision center capacity, Actor capacity, Actor strategy | Sensitivity of Decision Center to efforts to support them |

**Table 1.** Summary of modeled processes, the variables or sub-processes linked through these processes, and example scale or exponent parameters associated with the processes. The scale parameters represent the significance of certain processes in driving changes to the state variables, while the exponent parameters represent the sensitivity of the processes to the drivers. The last three processes are driven in part by actors' strategies, which are represented though how they allocate their capacity to maximize their resource access.





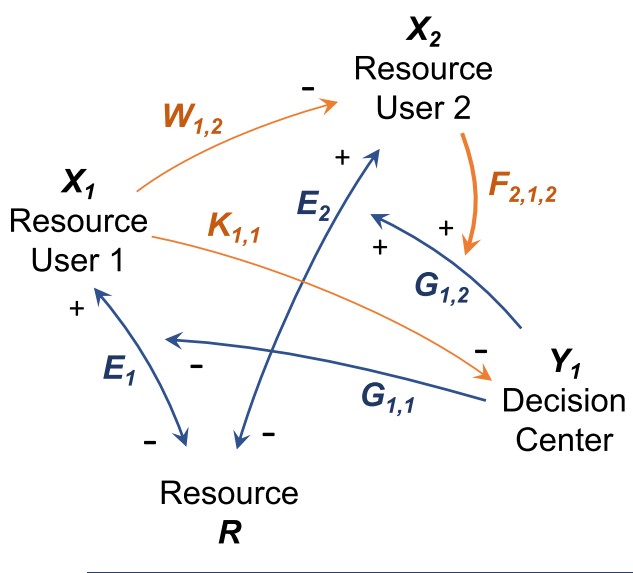

**Figure 1.** Example System Diagram. The nodes ($R$, $X_1$, $X_2$, and $Y_1$) are the state variables in the model, while the linkages represent functions (in blue) or parameters (orange) describing how the variables interact. In this example system, both resource users are extractors; Resource User 1's extraction is being restricted by the Decision Center ($G_{1,1}$), while Resource User 2's extraction benefits from intervention by the Decision Center ($G_{1,2}$). The orange linkages represent possible Nash Equilibrium strategies that may result from this setup. In this example, Resource User 2 puts all of their effort into supporting the intervention that increases their extraction ($F_{2,1,1}$), while Resource User 1 splits their effort between undermining the organizational capacity of Resource User 2 ($W_{1,2}$) and the Decision Center ($K_{1,1}$).

**Resource**

The dynamics of the resource $R$ follows a differential equation of the form

$$\dot{R} = S(R) - \sum_n E_n(R, G_{1,n}, \ldots, G_{M,n}),$$

where $S$ represents the reproduction/recharge and $E_n$ the rate of loss from extraction/exploitation by resource user $n$, which is itself a function of $R$ and interventions (e.g. regulations, subsidies, infrastructure) $G_{m,n}$ by decision center $m$ to either support or reduce each user's extraction. The effect of the intervention

$$G_{m,n} = G_{m,n}(Y_m, F_{1,m,n}X_1, \ldots, F_{N,m,n}X_N)$$





is then a function of the capacity of corresponding decision center and of efforts $F_{k,m,n}X_k$ by each actor $k$ to influence policies or the enforcement of these policies (see the section on Actors' Strategies for more details). In Figure 1, $F_{1,2}$ is an example

of such an effort. These nested functions integrate the resource state, actors' ability to access the resource, decision centers' efforts to intervene in their access, and, in turn, actors' efforts to influence these interventions.

**Resource Users**

The resource users' organizational capacity $X_n$ is modeled by

$$
\begin{aligned}
\dot{X}_n = {} & B_n(E_n(R, G_{1,n}, \ldots, G_{M,n})) \\
& + Q_n(A_n(R, P_{1,n}, \ldots, P_{M,n})) \\
& + \sum_k C^+_{k,n}(W^+_{k,n}X_k) - \sum_k C^-_{k,n}(W^-_{k,n}X_k) - L_n(X_n),
\end{aligned}
$$

where $B_n$ represents user $n$'s gain in capacity motivated by their ability to extract $E_n$. The function $Q_n$ is the analogous gain in capacity based on their non-extractive access to the resource, $A_n$. Depending on how these relationships to the resource are parameterized, these gain terms can represent different responses to resource access. This gain can represent actors becoming

more agitated due to lack of access to the resource, and thus more motivated to dedicate time and resources towards engaging with the institutions that determine their access. It can also represent actors becoming more invested in ensuring access to the resource as their use of the resource, and the value associated with it, increases. This parameter therefore encapsulates the importance of the resource to the users and the productivity of the system in determining their likelihood to self-organize, as described by Ostrom's social-ecological systems framework (Ostrom, 2009). The function $P_{m,n}$ is analogous to $G_{m,n}$,

representing interventions in their resource access and similarly affected by actors' efforts $H_{k,m,n}X_k$.

$C^+_{k,n}$ represents their gain in capacity from collaboration with or support from other actors that may, for example, provide information or resources about the institutions affecting their resource access or help connect them to these institutions (Barnes and van Laerhoven, 2015). They experience loss in capacity based on other actors' efforts to undermine them ($C^-_{k,n}$), through, for example, intimidation, misinformation, or demobilizing messaging and framing ($W_{1,2}$ in Figure 1). Actors also experience

a loss in capacity from attrition or turnover ($L_n$) due to a gradual loss in interest and participation among actors, or their switching attention to issues external to the model domain.

**Non-Government Organizations**

Non-government organizations includes non-profits, outreach, advocacy organizations, and other non-government organizations that typically are more public-facing and formal than resource user groups and may receive funding or grants from exter-

nal actors (e.g. donations or grants). These organizations play an important role in fostering and supporting collective action (Dobbin, 2021; Barnes et al., 2016; Barnes and van Laerhoven, 2015). non-government organizations are modeled similarly to


resource users by an equation of the form

$$\dot{X}_n = U_n(X_n) + \sum_k C_{k,n}^+(W_{k,n}^+ X_k) - \sum_k C_{k,n}^-(W_{k,n}^- X_k) - L_n(X_n),$$

where $U_n$ represents a gain in capacity from external sources.

Like resource users, these organizations have an objective related to the resource and are strategic, but have a gain term based on their own capacity, which allows them to secure external support, and do not have a gain term dependent on the resource, reflecting their more established and less reactionary nature.

**Decision Centers**

Decision centers, which can also be thought of as public infrastructure providers or venues for decisionmaking, intervene in resource users' ability to extract or access the resource, whether through provision of infrastructure, funding, or regulation. Their capacity $Y_m$ is modeled by

$$\dot{Y}_m = I_m^+(Y_m, K_{1,m}^+ X_1, \dots, K_{N,m}^+ X_N) - I_m^-(Y_m, K_{1,m}^- X_1, \dots, K_{N,m}^- X_N),$$

where $I_m^+$ represents their gain in capacity and $I_m^-$ represents their loss in capacity. Both of these terms are functions of a decision center's existing capacity as well as actors' efforts at venue shopping, in which actors attempt to move decision-making authority to venues that are more favorable to them These efforts are represented by $K_{k,m}^+ X_k$ for supporting a venue, and $K_{k,m}^- X_k$, to undermine a venue ($K_{1,1}$ in Figure 1).

**Actors' Effort Allocation**

Recognizing that actors in a governance system are strategic and self-organized, a quality that is unique among the complex systems for which stability has been studied in a systematic manner, the generalized model is coupled with an agent-based modeling component. Each actor allocates their limited organizational capacity among different actions in order to maximize their equilibrium extraction or access to the resource (or for non-government organizations, the access or extraction of another actor). Their strategies are thus subject to the constraint:

$$\left( \sum_k \sum_m |F_{n,m,k}| + |H_{n,m,k}| \right) + \left( \sum_k |W_{n,k}| \right) + \left( \sum_m |K_{n,m}| \right) = 1$$

where $F$, $H$, $W$, and $K$ represent the proportion of their effort dedicated to lobbying or otherwise directly supporting or resisting policies, collaborating with or undermining other actors, or directly influencing the capacity of a decision center, respectively. In Figure 1, for example, Resource User 1 is dividing their effort between undermining Resource User 2's capacity ($W_{1,2}$) and undermining the capacity of the decision center that is regulating them ($K_{1,1}$). The way actors allocate their effort among these actions is their strategy, which is calculated by finding a Nash equilibrium, in which no actor will want to unilaterally change their strategy.

While the generalized modeling approach means that the equilibrium extraction or access cannot be computed, the gradient of their extraction or access at the steady state can be calculated. As a simplified example of the method, take $\dot{X} = F(X, p)$,





where $p$ is a strategy parameter. To find how the steady state $X^*$ depends on the strategy parameter, we can write this equation

at the steady state with $X^*$ as a function of $p$:

$$0 = F(X^*(p), p).$$

Taking the total derivative of both sides with respect to the strategy parameter gives

$$0 = \frac{\partial F}{\partial X}\frac{dX^*}{dp} + \frac{\partial F}{\partial p}.$$

We can then solve for:

$$\frac{dX^*}{dp} = -\left(\frac{\partial F}{\partial X}\right)^{-1}\frac{\partial F}{\partial p}.$$

In the full system, $\left(\frac{\partial F}{\partial X}\right)^{-1}$ is the inverse of the Jacobian. Once we know how the steady state depends on the strategy parameters, it is straightforward to compute how the extraction or access depends on each strategy parameter.

A Nash equilibrium is calculated by computing the gradient of the equilibrium extraction or resource access and performing iterative steps of gradient descent for each actor in turn until the strategies converge.

**Generalized Modeling Approach**

In typical dynamical systems analysis, the functions in the equations above would then be assigned specific functional forms. However, generalized modeling is based on the recognition that computing steady states is computationally expensive, while

determining stability around a given steady state is far less costly. Bypassing the need for functional forms is particularly useful in modeling social systems, where the functional forms of processes are difficult to quantify and highly uncertain. This approach allows for analyzing system with a great degree of generality and without the computational constraints involved in modeling many different specific dynamical systems, and has been used to analyze a wide variety of systems (Gross et al., 2009; Gross and Feudel, 2006; Lade and Niiranen, 2017). Therefore, rather than assign specific functional forms and compute

the steady state, the functions are normalized by the unknown steady state. For example, the normalized resource dynamics would be represented by

$$\dot{r} = \frac{S^*}{R^*}s(r) - \sum_n \frac{E_n^*}{R^*}e_n(r, g_{1,n}, \ldots, g_{M,n})$$

where $S^*$, $R^*$, etc. are the values of the corresponding functions or state variables at equilibrium, and $s$, $x_k$, etc. represent the normalized functions or state variables. The normalization leads to the introduction of unknown factors $S^*/R^*$ and $E_n^*/R^*$.

However, these factors are constants and are treated as parameters, namely scale parameters. Scale parameters denote the magnitude of fluxes, such as turnover rates or the relative importance of different processes, while (Figure 1). We define $\phi := S^*/R^* = \sum_n E_n^*/R^*$, which represents the overall turnover rate of the resource, and $\psi_n := (1/\phi)(E_n^*/R^*)$, which represents the fraction of losses by each particular extractor at the steady state. The normalized resource dynamics can then be written as

$$\dot{r} = \phi\left(s(r) - \sum_n \psi_n e_n(r, g_{1,n}, \ldots, g_{M,n})\right).$$



An example of an entry of the Jacobian based on this equation can then be computed as

$$\frac{\partial \dot{r}}{\partial r} = \phi \left( \frac{\partial s}{\partial r} - \sum_n \psi_n \frac{\partial e_n}{\partial r} \right).$$

The derivatives $\partial s/\partial r$ and $\partial e_n/\partial r$ are unknowns that are also treated as parameters, namely exponent parameters. These parameters are an indication of the sensitivity of the growth rate and the extraction rate to the resource state, respectively. In general, exponent parameters indicate the non-linearity of a process at equilibrium. Once the Jacobian is parameterized, the

stability can be determined by checking whether the real part of all eigenvalues is less than 0.

A full description of all model parameters is given in Supplemental Table 1. These parameters collectively provide all of the information needed to compute the stability of the system. Varying the generalized modeling parameters, as well as meta-parameters defining the number of each type of state variable and how densely connected they are, allows for exploring the stability of a wide variety of topological configurations and feedbacks among actors and decision centers.

**Experimental Methods**

We investigate how three types of features of complex governance correspond with stability: 1) the scale and exponent parameters, to understand the importance of different processes and functional forms, as well as the importance of variance in these processes, representing, for example, heterogeneity in actors' response to resource access conditions, diversity in interventions, or inequity in actors' abilities to influence different governance processes, 2) the diversity of the system, indicated by the total

number of actors and decision centers, and how densely connected they are, and 3) the relative number of decision centers and actors, to understand, for example, whether diverse stakeholder interests may have a different effect on stability than diversity in decision centers.

**Parameter Correlations with Stability**

To understand the impact of the scale and exponent parameters on stability, the size and composition of the system is held fixed

and the scale and exponent parameters are sampled independently (see Supplementary Information for the parameter ranges). The small system has a total size of 5, with one extractor, one accessor, one non-resource user actor, and two decision centers. The large system has a total size of 15, with three of each type of resource user (extractors, accessors, and combined extractors and accessors), three non-resource user actors, and three decision centers. The decision center-resource user connectance (i.e., the probability that a given decision center will intervene in a particular resource user's extraction or access) is fixed at 0.4.

The experiment consisted of 28,800 samples. The sample size was chosen to sufficiently narrow the 95% confidence intervals so that statistically significant correlations were distinguishable.

Stability is treated as a binary value. The correlation of a given parameter, $x$, averaged across all actors, with stability is given by

$$R = \frac{\sum_{i=1}^{v_s} x_{s,i} - \frac{v_s}{v} \sum_{i=1}^{v} x_i}{v \sigma_x \sigma_s},$$



where $x_{s,i}$ is the set of parameter values leading to stable systems, $x_i$ is the ensemble of parameter values, $v$ is the number of systems in the ensemble, $\sigma_x$ is the standard deviation of the parameter values, and $\sigma_s$ is the standard deviation of stability (0 for unstable systems, 1 for stable systems).

**Colormaps**

To explore the effect of polycentricity or diversity of actors, the exponent parameters are held fixed and assigned the values
in Supplementary Table 1 to eliminate variation across exponent parameters as the decision center-resource user connectance and the total size of the system is varied. Connectance represents the proportion of possible interactions in the network that are realized. Since the final system connectance is determined by the strategies actors pursue, it is computed after optimization of actors' strategies rather than fixed a priori.

For each system, the composition is randomly sampled, with a minimum of two resource users, at least one of which is
an extractor, and one decision center. In the ternary colormaps (Figure 3), the total size of the system is held fixed at 10, and the decision center-resource user connectance is held fixed at 0.4. There are 600 systems sampled for each combination of connectance and size, and 900 for each system composition in the ternary colormaps. Sample sizes were chosen such that additional samples do not significantly change the trends in the colormaps.

**Results and Discussion**

**Parameter Correlations with Stability**

The parameter correlations results reveal that for a smaller system (Figure 2a), stabilizing factors include greater capacity gains from external support and gains motivated by resource access conditions, as opposed to gains from collaboration, and greater losses based on attrition rather than being undermined. Conversely, a greater capacity gain from collaboration and capacity loss from undermining by other actors, as well as a greater proportion of actors' efforts spent on collaboration or undermining are
destabilizing. This suggests that stronger interactions and greater interdependence among actors is destabilizing, while greater autonomy is stabilizing.

In both the smaller and larger systems (Figure 2b), the strategy parameters emerge as important in determining stability. Since the strategies are computed rather than sampled as the other parameters are, the causal effect of the strategies on stability is not clear. However, the results suggest that a greater effort put toward influencing the capacity of decision centers, or venue
shopping, corresponds with stability, while greater effort put into the other strategies corresponds with reduced stability. Venue shopping has gained interest as a possible mechanism through which less powerful actors can enact fundamental policy changes (Pralle, 2003). This distinction suggests that venue shopping may arise as a desirable strategy in a different context than other political strategies, or that it changes the system in a fundamentally different manner from other strategies, and that it does so in a lasting manner.





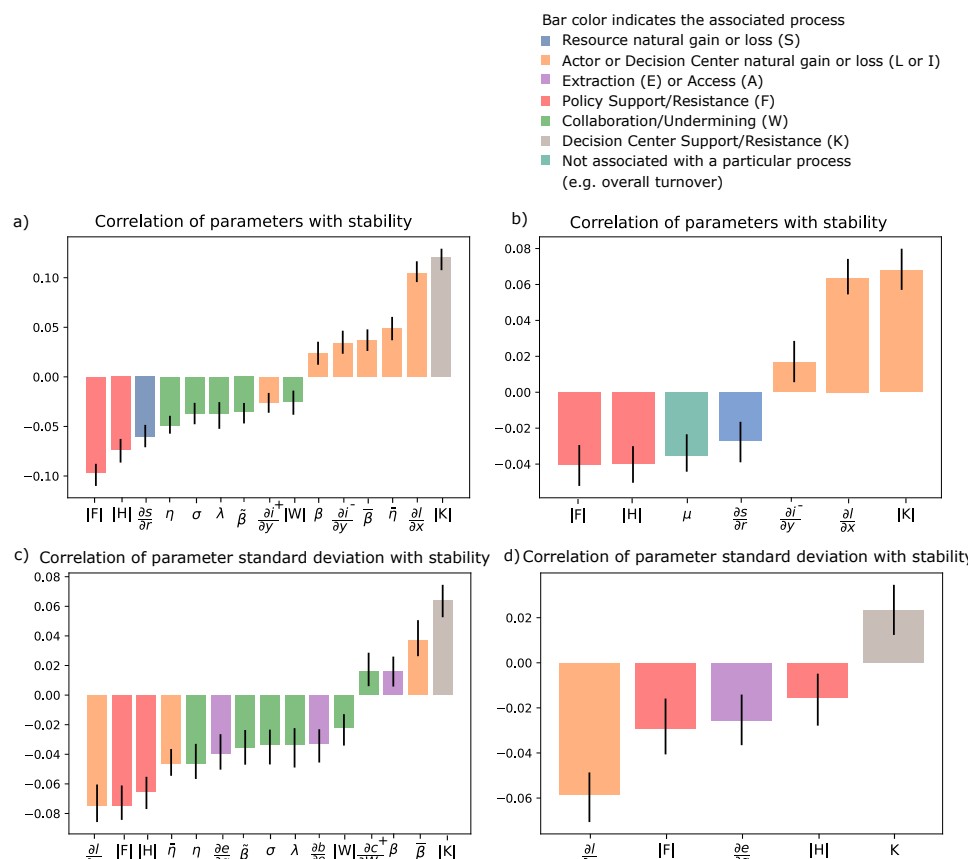

**Figure 2.** Correlation of mean (top) and standard deviation (bottom) of parameters with stability in small (left) and large (right) systems. Only parameters with a statistically significant effect on stability and with a correlation greater than 0.01 are shown. Stabilizing factors include the proportion of effort put into influencing the capacity of decision centers ($|K|$), sensitivity of attrition to the current organizational capacity ($\partial l/\partial x$), the share of loss in capacity due to attrition ($\overline{\eta}$), share of gain in capacity from organization self-growth efforts ($\overline{\beta}$), the sensitivity of decision center loss in capacity to their capacity ($\partial i/\partial y_n$), and the gain in capacity motivated by resource access conditions ($\beta$). Destabilizing factors consist of the proportion of effort put into spent on influencing effectiveness of policies ($|F|$ and $|H|$) and on collaborating or undermining ($|W|$), the sensitivity of resource regeneration to the resource state ($\partial s/\partial r$), share of loss in capacity from undermining by other actors ($\eta$ and $\lambda$), share of gain from collaboration ($\widetilde{\beta}$ and $\sigma$), and the sensitivity of decision center growth in capacity to their own capacity ($\partial i/\partial y_p$). The standard deviation results reveal that in addition to these parameters, variation in the sensitivity of extraction to the intervention ($\partial e/\partial g$) and the sensitivity of gain in capacity to the ability to extract ($\partial b/\partial e$) is destabilizing.

To understand the effect of heterogeneity among actors' relationships to the resource or to institutions on stability, we look at the variation in the parameters that define, for example, their sensitivity to changes in resource accessibility, or their share





of total resource extraction. We find that higher variation in the sensitivity of actors' extraction to governance is destabilizing. This variation corresponds with heterogeneity among resource users in terms of the ease with which their extraction or access can be monitored or regulated. It can also represent institutional diversity, in which decision centers pursue a variety of policies or approaches, which has been hypothesized to increase adaptive capacity (Carlisle and Gruby, 2019). A higher variation in the sensitivity of actors' gain in organizing capacity to their resource access is also destabilizing. Differences in this parameter correspond to different relationships with resource use: actors with low resource requirements, particularly if they are not involved in a profit-driven activity, may experience the largest capacity gains when their ability to extract is low. In contrast, some actors may become more invested and gain greater resources with which to mobilize as their extraction increases. The presence of both of these relationships to the resource similarly signify heterogeneity and potentially inequity among actors, leading to a greater tendency for contestation and change.

**Effect of Polycentrism and Diversity on Stability**

The number of different groups in the system, whether actors or decision centers, has a strong effect on stability, while connectance has no noticeable effect on stability (Figure 3). This suggests that diversity in actors, a feature of complex and polycentric governance systems, is a destabilizing force. This is consistent with the idea that the inclusion of a greater diversity of actors in governance processes leads to greater flexibility and adaptability, as well as with findings for other complex systems such as ecosystems (May, 1972). The absence of an effect of connectance on stability, however, is in contrast to other complex systems where connectance is destabilizing (May, 1972). This may be because each of the interactions in these systems can influence processes by either increasing or decreasing their effect, unlike in natural systems, where interactions may all push the system in the same direction, leading to greater potential for destabilizing feedbacks. Thus, while greater connectivity in the form of stronger interactions is a destabilizing force, as found in the parameter correlation experiments (Figure 2), the presence or absence of interactions is not as important for determining stability in governance systems.



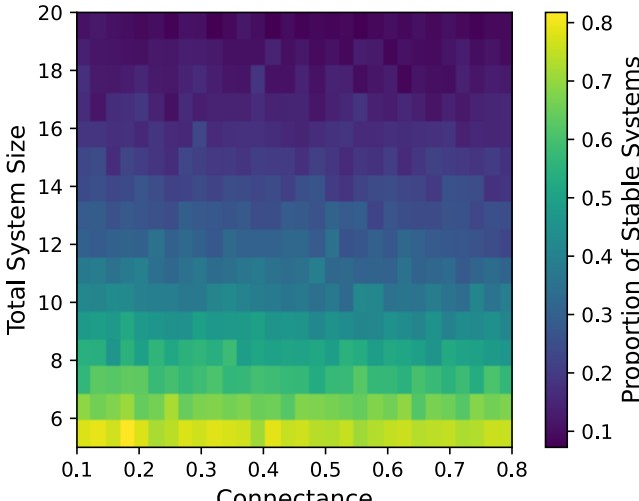

**Figure 3.** Effect of system size (number of actors and decision centers) and connectance on stability. The color represents the proportion of stable systems (out of 600 samples) for each connectance and system size. The connectance shown here is the proportion of links between decision centers and resource users' extraction or access ($G_{1,1}$ and $G_{1,2}$ in Figure 1); the same result holds for the total connectance as well (see Figure S2).





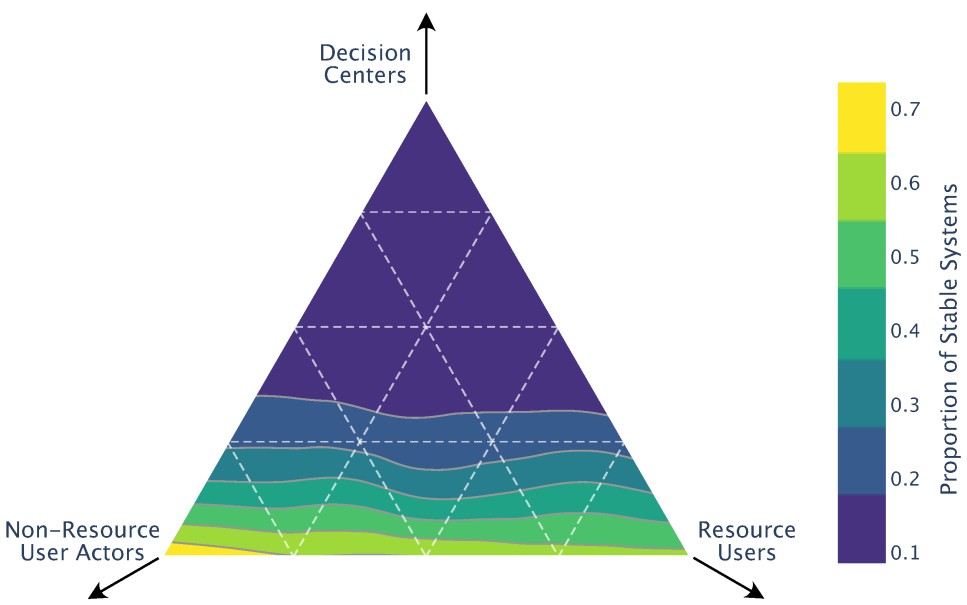

**Figure 4.** Effect of the number of resource users, non-government organizations, and decision centers on stability. The color represents the proportion of stable systems for a given system composition. The total system size is 10, with a minimum of 2 resource users and 1 decision center.

**Effect of System Composition on Stability**

Finally, looking at the effect of the relative proportions of different entities reveals that stability is determined not just by total diversity, but the diversity in decision centers in particular(Figure 3). A greater number of decision centers is destabilizing, while a greater proportion of non-government organizations is stabilizing. The proportion of resource users does not have a strong effect on stability. Whether the resource users are extractive users of the resource also does not have an effect on stability (Figure S3). This result thus supports that polycentrism causes governance systems to be more prone to change, likely because they offer more opportunities for actors to influence the system (Pralle, 2003). Non-government organizations may have a stabilizing effect because of their role in supporting, and thus having aligning goals, with other actors in the system, reducing contestation and helping other actors develop longer-lasting, more durable institutions (Barnes and van Laerhoven, 2015).

**3  Conclusions**

In this study, we propose a modeling framework for resource governance that couples dynamical systems modeling with an agent based model representing actors' strategic interactions with each other and the institutions and organizations mediating their access to a shared resource. By formulating this system as a generalized model,we are able to explore a variety of structures for these relationships, with varying system compositions, types of relationships, connectances, and sizes to identify





the factors that influence stability. This approach reveals that greater interdependence and heterogeneity in actors' responses to resource access conditions, as well as in the institutions affecting their resource access, are destabilizing. Additionally, a greater number of different entities, especially a greater number of decision centers, is destabilizing, while greater diversity in
non-government organizations is stabilizing. Finally, the strategy of venue shopping corresponds with stability, while strategies such as supporting or undermining other actors or policies correspond with instability.

The applicability of the results are ultimately contingent on whether the modeled processes, such as actor's attempts to navigate governance and support or resist institutions to increase their resource access, are indeed the driving forces in the governance system. Therefore, they may not necessarily apply to governance driven mainly by top-down bureaucratic processes
with little stakeholder engagement or with very high capacity to monitor, implement, and enforce policies. Additionally, actors are represented as rational in the model, rather than the often heuristic and myopic manner in which they actually form their strategies for navigating governance (Pralle, 2003).

Despite these limitations, this study provides new insight into the factors that determine how governance systems respond to change, as well as independent support for previously observed benefits of complex governance. Many of the factors com-
monly associated with complex governance, namely greater interdependence and diversity in actors and decision centers, are destabilizing. This suggests that, similar to other complex systems, complexity in governance systems is destabilizing (May, 1972). It may be this courting of instability that allows for complex governance to be more responsive to external change (Zumsande et al., 2011). However, some results, such as the lack of effect of connectance on stability, contrast with findings for ecosystems, while the stabilizing effect of factors such as a greater number of non-government organizations and venue
shopping have not previously been explored systematically. These differences suggest that there is a benefit to modeling the dynamics of governance systems specifically, rather than extending ecological theories to social systems. These results also suggest some concrete strategies to strike a balance between adaptivity and extreme instability in complex governance by, for example, introducing mitigating factors like non-government organizations to help stabilize systems with many different actors.

While modeling is not a replacement for case studies in understanding complex governance, it is complementary by suggesting new theories, such as the stabilizing effect of non-government organizations or of venue shopping, along with providing more detailed insight into existing theories. These results, for example, provide greater insight into the greater adaptivity of polycentric governance by elucidating which factors – such as a greater number and diversity of decision centers as opposed to all entities, and greater interdependence among actors rather than simply the density of connections – lead to greater instability.
As suggested by numerous studies (McGinnis and Ostrom, 2012; Stephan et al., 2019; Thiel et al., 2019; Carlisle and Gruby, 2019), this level of detail is necessary in understanding when or why the many benefits ascribed to multi-level, complex, and polycentric systems actually materialize. While there remains a need for more systematic exploration of other system-level outcomes in addition to how governance systems respond to change, this study demonstrates a way forward in applying the insights of complex systems theory to managing complex and highly uncertain human-natural systems in the face of rapid
social and environmental change.



*Code availability.* All code used in this study is available at https://github.com/njmolla/Gen-Modeling-Governance.

*Author contributions.* N.M., J.D., and T.G. developed the model, N.M. designed and conducted experiments, and N.M. wrote the paper with feedback from J.D., T.G., and J.H.

*Competing interests.* The authors declare that they have no competing interests.

*Acknowledgements.* NM received support from the Gates Millennium Scholars Program. JH received partial support from the U.S. National Science Foundation, grant CNH-1716130. Conclusions are those of the authors and do not necessarily reflect the views or policies of the NSF.





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
