# Peer review of "Governing Change: A Dynamical Systems Approach to Understanding the Stability of Environmental Governance"

_Earth System Dynamics, 2022_

## Author Comment (AC2)

**Response to Reviewer 1**

*This manuscript provides a very interesting and innovative set of stylized modeling experiments to explore the stability of governance structures in environmental systems. The evaluation of system stability across thousands of governance structures using a generalized dynamic systems modeling approach is particularly novel and insightful. The manuscript is well written and organized. I'd suggest the authors address the following comments to further improve the manuscript:*

*1. The modeling approach necessarily deals with a stylized, abstracted representation of environmental governance systems. While there is some attempt to draw analogies between the mathematical abstraction and real-world systems in the introductory text, such analogies are largely excluded from the description of the model itself, the results, and the discussion/conclusion. I think the manuscript could be improved by providing examples of tangible aspects of real-world systems that the mathematical abstractions might represent (or using a single example, e.g., groundwater systems, and carrying it through the entire manuscript to aid readers with interpretability and bring the modeling formulation to life a bit more)*

We have revised Figure 1 and the caption to provide a more concrete example system that is referred to throughout the paper:

| Example Parameter | Type | Meaning |
|---|---|---|
| $\dfrac{\sum_n E_n}{R}$ | Scale | Extraction rate of resource (i.e. inverse of characteristic time scale of resource) |
| $\dfrac{E_1}{E_1 + E_2}$ | Scale | Share of total extraction extracted by Resource User 1 |
| $\dfrac{\partial E_1}{\partial R}$ | Exponent | Sensitivity of Resource User 1's extraction to resource level |

[Figure]

**Example System Diagram.** The nodes ($R$, $X_1$, $X_2$, and $Y_1$) are the state variables in the model, while the linkages represent functions (in blue) or parameters (orange) describing how the

variables interact. In this example water governance system, there are two types of water users, agricultural users and urban users, withdrawing water from a reservoir. The governance intervention $G_{1,1}$ in this example can be interpreted as infrastructure managed by the infrastructure provider, or Decision Center, that delivers water to the city, supporting urban extraction while reducing agricultural extraction. The orange linkages represent possible Nash Equilibrium strategies that may result from this setup. In this example, urban users allocate their effort to supporting the infrastructure that allows for their extraction ($F_{2,1,2}$), while agricultural users split their effort between undermining the organizational capacity of urban users ($W_{1,2}$) and of the Decision Center ($K_{1,1}$).

We have added the following references to this example system throughout the Modeling Approach section:

In the example system, $S$ represents the natural net gain to the reservoir after natural inflows and outflows that are not delivered to any users, and $E_1$ the total amount that agricultural users are able to extract.

In Figure 1, $F_{1,2}$ is an example of such an effort that could represent urban users advocating for increasing the conveyance efficiency of the infrastructure delivering their water.

These efforts are represented by $K_{k,m}^+ X_k$ for supporting a venue, and $K_{k,m}^- X_k$, to undermine a venue. $K_{1,1}$ in Figure 1 for example, may represent agricultural users' efforts to undermine the authority of the Infrastructure Provider to withdraw water to deliver to urban water users.

In Figure 1, for example, farmers divide their effort between undermining urban users' capacity ($W_{1,2}$) and undermining the capacity of the infrastructure provider that conveys water to urban users and away from farmers ($K_{1,1}$).

We have also added the following to the Results and Discussion section aid interpretation of the results:

However, the results suggest that a greater effort put toward influencing the capacity of decision centers, or venue shopping, corresponds with stability, while greater effort put into the other strategies corresponds with reduced stability. In the example system, agricultural users are engaging in venue shopping by reducing the infrastructure provider's influence over the infrastructure ($K_{1,1}$); if there were other decision centers in the system, they may try to move that authority to a venue favors agricultural interests.

Differences in this parameter correspond to different relationships with resource use: actors with low resource requirements, particularly if they are not involved in a profit-driven activity, may experience the largest capacity gains when their ability to extract is low. In contrast, some actors may become more invested and gain greater resources with which to mobilize as their extraction increases. In the example system, for example, urban interests will likely become less engaged once they have sufficient access to water (an inverse relationship between capacity and resource access), whereas agricultural users, particularly those part of industrial agriculture operations, might become less engaged once the available water, and thus profitability of farming, drops below a certain threshold.

2. *I'm not seeing where "stability" is clearly defined, both conceptually and mathematically. Perhaps I missed it. Regardless, a concise definition of the concept should be up front and center given the manuscript's focus.*

We have made the following changes to clarify the conceptual and mathematical meaning of stability and make those definitions more prominent:

In the introduction, we have moved the definition of stability earlier and added the mathematical definition in addition to the conceptual definition:

> Given that constant change is a central feature of complex systems, a system-level outcome of particular interest is stability. Mathematically, a steady state with local asymptotic stability is one for which trajectories near the steady state will approach the steady state. Conceptually, local asymptotic stability, hereafter referred to as stability, is an indication of the system's ability to retain its structure and function in the face of local perturbations in the variables controlled by the governance system (Guckenheimer and Holmes, 1983).

In the Modeling Approach section, we have revised the "Generalized Modeling Approach" heading to "Generalized Modeling Approach to Computing Stability" and have added the following text:

> Once the Jacobian is parameterized, the stability can be determined by checking whether the real part of all eigenvalues is negative. Conceptually, this means that perturbations in the state variables close to the steady state will return to that steady state. Local stability therefore indicates that the system will return to a steady state under short-term shocks (e.g. a sudden change to an actor's political influence), but does not necessarily indicate how the system will respond to large perturbations from the steady state or long-term drivers that fundamentally change the system's functioning (e.g. altering how resource users benefit from or impact the resource).

3. *While the effort to deploy thousands of structural variants of the environmental system is impressive and laudable, it seems to me that the revealed system dynamics still may be subject to higher-level structural assumptions regarding the nature of actor interactions. For example, NGOs are not directly tied to the state of the resource, whereas one might argue that NGOs are inversely (loss term rather than gain term) related to resource state (e.g., the tendency for environmental NGOs to emerge/grow as a particular environmental resource degrades). Likewise, actors could be viewed as operating within a nested structure (e.g., individual resource users interested in preservation of a resource comprising an NGO). While I understand that such a stylized formulation cannot touch upon all of these elements, I think the advantages/disadvantages of the proposed formulation can be further interrogated in the discussion.*

It is true that even with the Generalized Modeling formulation, some structural assumptions are inevitable.

We have added the following discussion of these types of higher-level modeling assumptions in the Conclusion section:

> Additionally, even though the generalized modeling approach requires fewer assumptions than traditional dynamical systems analysis, there are still assumptions regarding the structure of interactions among different model components. For example, the change in capacity of non-government organizations and decision centers does not directly depend on the resource state, but rather is affected by the resource state only indirectly through its influence on resource users' capacities and actions. Ultimately, we aimed to achieve a balance between a more general model that would make few assumptions about the structure of interactions, but would be challenging to interpret in the context of resource governance systems, and a more structured model, which limits

the variety of ways in which variables are linked but provides more precise insight into governance dynamics.

In the NGO case, the assumption is that if NGOs grow or decline based on the state of an environmental resource, it is due to support/lack of support from resource users in the system. This does not account for the fact that NGOs may have more trouble obtaining external sources such as grants or donations or may have more trouble recruiting staff when a resource is no longer threatened and vice versa since the assumption is that external forces will be less reactionary than resource users that are directly impacted by a resource. In a similar vein, regarding actors operating within a nested structure, while the model does not explicitly represent individual resource users comprising an NGO, it would represent a group of resource users collectively working with an NGO as a collaborative relationship in the model, where resource users help the NGO grow in capacity and their support depends on the state of the resource.

4.  *Given the modeling interest on actors' ability to influence policies or capacities of other actors, there might be some interesting and relevant connections with the power relations and sustainability transitions literature (see for example Avelino and Wittmayer, 2016). Perhaps this could be further explored in the introduction and/or discussion/conslusion.*

Thank you for the suggestion, the power relations and sustainability transitions literature has clear connections with the concepts underlying the model. The following references have been added to the Modeling Approach section:

> This bottom-up perspective is chosen because of the under-representation of actors' agency in making and influencing decisions and pursuing their goals in the polycentric governance literature, which tends to focus solely on structure and exclude entities that lack the authority to create policies, though this is changing with concepts like institutional navigation (Dobbin, 2021; Villamayor-Tomas and García-López, 2018) and the sustainability transitions literature, which emphasize actors and the dynamic power relations among them as a driving force behind governance transitions (Avelino and Wittmayer, 2016}.

And to the conclusion:

> Additionally, while this study focuses on analyzing theoretical systems, the ability to model the different ways that actors exercise power and the dynamic power relations among them allows for exploring questions relating to the interaction between governance transitions and power relations in empirical systems as well (Avelino and Wittmayer, 2016; Avelino, 2021). This study demonstrates a way forward in combining the insights of complex systems theory with theories on governance to managing complex and highly uncertain human-natural systems in the face of rapid social and environmental change.

5.  *The assumption of a Nash-equilibrium in actors' allocation of efforts is a strong one and receives very limited treatment in the manuscript. While I understand the adoption of the approach from a computational and conceptual standpoint, I think further elucidation of the implications and limitations of such an approach is warranted.*

This is a good point. Given that the strategy space is not necessarily convex, there is no guarantee of a Nash equilibrium. However, the modeling approach does not rely on the existence of a Nash equilibrium. To give results that are meaningful for understanding governance systems, the model only requires that actors behave in ways that are feasible (i.e. actions that don't contradict themselves) and are in their selfinterest, which the optimization method does ensure. We have added the following to the Modeling Approach section to clarify this point:

> A Nash equilibrium is calculated by computing the gradient of the equilibrium extraction or resource access and performing iterative steps of gradient descent for each actor in turn until the strategies converge. While there is no guarantee of a Nash equilibrium since the strategy space is not necessarily convex, the strategy optimization process ensures that even if optimality is not reached, actors are behaving in ways that are self-consistent and compatible with their goal of increasing their resource access. Modeling actors as behaving reasonably, if not necessarily rationally, ensures that the systems that are analyzed are feasible governance systems.

In addition, to address the concern as to whether a Nash equilibrium is a realistic representation of actor's strategies, we have added the following discussion to the conclusion:

> Additionally, the model assumes a Nash equilibrium in actors' strategies, representing actors as rational and having perfect knowledge of the system and others' actions, rather than the often heuristic and myopic manner in which they actually form their strategies for navigating governance (Pralle, 2003). However, this assumption is more reasonable in stable systems, where repeated interactions in a stable environment allow actors' greater opportunity to learn about the system and fine-tune their strategies (Craig et al., 2017; Pahl-Wostl, 2009).

6. *The abstract mentions a system's ability to "adapt to social and environmental change" and recover from "perturbations". Can the authors speak more to how perturbations of the system (in the form of either short-term shocks or gradual stressors factors) relates to the formulation? What exactly are the "perturbations in the variables controlled by the governance system" in this particular setup? And how does the concept of stability connect? I think a clearer defintion of stability (see comment above) and some added discussion could bring clarity to this.*

The paper uses the concept of a system's ability to recover from perturbations interchangeably with stability. In this setup, the "perturbations in the variables controlled by the governance system" refers to perturbations in variables such as actors' influences or resource state (i.e. state variables) as opposed to perturbations to parameters such as resource regeneration rates. The revisions in response to (2) above clarifies the connection between stability and system recovery to perturbations, as well as the types of perturbations that can be understood through local stability analysis.

7. *Minor editorial comments:*

*Figure 1 - mismatch between F2,1,1 in the legend and F2,1,2 on the figure*

Fixed.

*Line 126 - "non-government" to "Non-government"*

Fixed.

*Line 138 - "them These" to "them. These..."*

Fixed.

*Figure 2 - Add "small system" and "large system" labels to the graphs (not just the captions) for readability*

Figure 2 has been revised as suggested:

[Figure]

**Response to Reviewer 2**

*This manuscript is a very welcome interdisciplinary contribution to Earth System Dynamics at both methodological and applied levels. Methodologically, it brings out solid dynamical systems approaches to addressing the highly nontrivial problem of environmental governance, where natural and human processes and interactions come into play that require not only the traditional dynamical systems principles in a sterile manner, but also social systems thinking with active decision making rather than the classical determinism. In this regard, this is a very insightful contribution that finds good grounds in an emerging but already reliable literature at the interface between natural and social systems with robust analytical mechanics principles and metrics (and dynamical systems in particular).*

*The stylised nature of the mathematical conceptualisations and experiments is crucial to shed light onto key interactions, with neither aiming at too much detail, nor at a too-macro of a picture that would wash out critical nonlinearities. As such, this is a very well balanced study, obviously with the inherent limitations that come with such exercise. The authors have done a pretty good job in laying*

*down their reasoning so that it is clearly understood where things come from and what they are meant to represent.*

*However, it is important to further clarify to those readership that is perhaps not so familar with one of either dynamical systems or governance reasoning the key notions being applied since aspects such as stability per se mean different things to different scientific communities. Further mathematical detail, while often discouraged in other venues, is never too much in this study, hence the authors are encouraged to add, perhaps in annex not to break the pleasant and clear flow of the text, further details on the underlying mathematical physics principles supporting their formal reasoning and formulation.*

In order to clarify the meaning of stability, we have made the following revisions:

In the introduction, we have moved the definition of stability earlier and added the mathematical definition in addition to the conceptual definition:

> Given that constant change is a central feature of complex systems, a system-level outcome of particular interest is stability. Mathematically, a steady state that has local asymptotic stability is one for which trajectories near the steady state will approach the steady state. Conceptually, local asymptotic stability, hereinafter referred to as just stability, is an indication of the system's ability to retain its structure and function in the face of local perturbations in the variables controlled by the governance system (Guckenheimer and Holmes, 1983).

In the Modeling Approach section, we have revised the "Generalized Modeling Approach" heading to "Generalized Modeling Approach to Computing Stability" and have added the following text:

> Once the Jacobian is parameterized, the stability can be determined by checking whether the real part of all eigenvalues is less than 0. Conceptually, this means that perturbations in the state variables close to the steady state will return to that steady state. It is worth noting that local stability therefore indicates that the system will return to a steady state under short-term shocks to the steady state (e.g. a sudden change to an actor's political influence), but does not necessarily indicate how the system will respond to large perturbations from the steady state or long-term drivers that fundamentally change the system's functioning (e.g. altering how resource users benefit from or impact the resource).

While a full justification of the validity of the Generalized Modeling method is outside the scope of this paper, and we refer readers to Gross and Feudel, 2006 for this, we have expanded the Supplementary Information to include the full mathematical derivation of the generalized parameters and Jacobian, and the calculation of the objective function gradient. Please find the revised supplementary information attached.

*The conditions under which their formulations are applicable and not should also be further discussed with additional few sentences so that the more naive reader is not tempted to throw the models around without enough care. The authors were clearly careful and that is very well seen through the solidity of their argumentation, formulation, results and discussion. But an additional pedagogic little touch would be the cherry on top of the cake to further help the increasinly mathematically fragile geoscience readership and even more so those coming from the more social science side that might alsot be interested.*

We have made the following revisions to address the assumptions underlying the modeling approach:

We have added the following discussion of higher-level modeling assumptions in the Conclusion section:

> Additionally, even though the generalized modeling approach allows for making fewer assumptions than traditional dynamical systems analysis, there are still assumptions regarding the structure of interactions among different model components. For example, the change in capacity of non-government organizations and decision centers is modeled as not directly depending on the resource state, but rather being affected by the resource state only indirectly through how it changes resource users' capacities and actions. Ultimately, we aimed to achieve a balance between a more general model in which every variable can impact every other variable, making few assumptions about the structure of interactions but also limiting the insight into the dynamics specific to resource governance systems, and a more structured model, which limits the variety of ways in which variables are linked in the model.

We have added the following discussion about the assumptions underlying the search for a Nash Equilibrium to the Modeling Approach section:

> A Nash equilibrium is calculated by computing the gradient of the equilibrium extraction or resource access and performing iterative steps of gradient descent for each actor in turn until the strategies converge. While there is no guarantee of a Nash equilibrium since the strategy space is not necessarily convex, the strategy optimization process ensures that even if optimality is not reached, actors are behaving in ways that are self-consistent and compatible with their goal of increasing their resource access. Modeling actors as behaving reasonably, if not necessarily rationally, ensures that the systems that are analyzed are feasible governance systems.

In addition, to address the concern as to whether a Nash equilibrium is a realistic representation of actor's strategies, we have added the following discussion to the Conclusion:

> Additionally, the model assumes a Nash equilibrium in actors' strategies, representing actors as being rational and having perfect knowledge of the system and other's actions, rather than the often heuristic and myopic manner in which they actually form their strategies for navigating governance (Pralle, 2003}. However, this assumption is more reasonable in stable systems, where repeated interactions in a stable environment allow actors' greater opportunity to learn about the system and fine-tune their strategies (Craig et al., 2017; Pahl-Wostl, 2009).

Finally, to aid understanding of the abstract concepts in the model, we have added a concrete example system that is referred to throughout the Modeling Approach and Results and Discussion sections (see our response to Reviewer 1).

*Last but not least, the remarks raised by the other referee are also hereby endorsed and will not be repeated. I would not say better in such regards.*

*All in all, this manuscript is definitely suitable for publication at Earth System Dynamics, is mostly appropriate at the scientific and technical levels safe for the minor aspects raised by both of us, and would also benefit from providing an extra layer of clarification and caution so that a broader readership other than us more technically minded can actually appreciate better the value and harness the vast potential of this contribution.*

*Thank you and all the best.*

**Supplementary Information for Governing Change: A Dynamical Systems Approach to Understanding the Stability of Environmental Governance**

Nusrat Molla[1], John DeIonno[2], Thilo Gross[3], Jonathan Herman[2]

[1]Land, Air, and Water Resources, University of California, Davis, Davis CA 95616

[2]Civil and Environmental Engineering, University of California, Davis, Davis, CA 95616

[3]Helmholtz Institute for Functional Marine Biology, University of Oldenburg, Oldenburg, Germany

**Contents**

**Supplementary Methods**

**Derivation of Generalized Modeling Scale Parameters**

We first define the state variables normalized by their steady state value ($R^*$ is the steady state value for $R$, for example):

$$r := \frac{R}{R^*}, \quad x_n := \frac{X_n}{X_n^*}, \quad y_m := \frac{Y_m}{Y_m^*}.$$

**Resource Equation**

We can then write the normalized functions

$$s(r) := \frac{S(rR^*)}{S^*}$$

and

$$e_n(r, g_{1,n}, \ldots, g_{M,n}) := \frac{E_n(rR^*, G_{1,n}^* g_{1,n}, \ldots, G_{M,n}^* g_{M,n})}{E_n^*},$$

where $g_{m,n}(y_m, F_{1,m,n} x_1, \ldots, F_{K,m,n} x_K) := \dfrac{G_{m,n}(y_m Y_m^*, F_{1,m,n} x_1 X_1^*, \ldots, F_{K,m,n} x_K X_K^*)}{G_{m,n}^*}.$

This allows us to rewrite the equation for $\dot{R}$ in terms of the normalized variables and functions:

$$\dot{r} = \frac{S^*}{R^*} s - \sum_n \frac{E_n^*}{R^*} e_n.$$

We then define the *scale parameters*

$$\phi := \frac{S^*}{R^*} = \sum_n \frac{E_n^*}{R^*}, \quad \psi_n := \frac{1}{\phi} \frac{E_n^*}{R^*},$$

and finally rewrite the normalized equation as

$$\dot{r} = \phi \left( s - \sum_n \psi_n e_n \right).$$

**Resource User and Non-Resource User Actor Equations**

We write the normalized functions

$$b_n(e_n) := \frac{B_n(E_n^* e_n)}{B_n^*}, \quad q_n(a_n) := \frac{Q_n(A_n^* a_n)}{Q_n^*},$$

$$c_{k,n}^+(W_{k,n}^+ x_k) := \frac{C_{k,n}(W_{k,n}^+ x_k)}{C_{k,n}^{+*}}, \quad c_{k,n}^-(W_{k,n}^- x_k) := \frac{C_{k,n}(W_{k,n}^- x_k)}{C_{k,n}^{-*}},$$

$$u_n(x_n) := \frac{U_n(x_n X_n^*)}{U_n^*}, \quad l_n(x_n) := \frac{L_n(x_n X_n^*)}{L_n^*}.$$

where $a_n(r, p_{1,n}, \ldots, p_{M,n})$ and $p_{m,n}(y_m, H_{1,m,n} x_1, \ldots, H_{K,m,n} x_K)$ are defined analogously to $e_n$ and $g_{m,n}$, respectively.

This allows us to rewrite the equation for $\dot{X}_n$ in terms of the normalized variables and functions:

$$\dot{x}_n = \frac{B_n^*}{X_n^*} b_n + \frac{Q_n^*}{X_n^*} q_n + \frac{U_n^*}{X_n^*} u_n + \sum_k \frac{C_{k,n}^{+*}}{X_n^*} c_{k,n}^+ - \sum_k \frac{C_{k,n}^{-*}}{X_n^*} c_{k,n}^- - \frac{L_n^*}{X_n^*} l_n.$$

We then define the *scale parameters*

$$\alpha_n := \frac{B_n^*}{X_n^*} + \frac{Q_n^*}{X_n^*} + \frac{U_n^*}{X_n^*} + \sum_k \frac{C_{k,n}^{+*}}{X_n^*} = \sum_k \frac{C_{k,n}^{-*}}{X_n^*} + \frac{L_n^*}{X_n^*},$$

$$\beta_n := \frac{1}{\alpha_n} \frac{B_n^*}{X_n^*}, \quad \widehat{\beta}_n := \frac{1}{\alpha_n} \frac{Q_n^*}{X_n^*}, \quad \overline{\beta}_n := \frac{1}{\alpha_n} \frac{U_n^*}{X_n^*}, \quad \widetilde{\beta}_n := \frac{1}{\alpha_n} \sum_k \frac{C_{k,n}^{+*}}{X_n^*}, \quad \sigma_{k,n} := \frac{1}{\alpha_n \widetilde{\beta}_n} \frac{C_{k,n}^{+*}}{X_n^*},$$

$$\overline{\eta}_n := \frac{1}{\alpha_n} \frac{L_n^*}{X_n^*}, \quad \eta_n := \frac{1}{\alpha_n} \sum_k \frac{C_{k,n}^{-*}}{X_n^*}, \quad \lambda_{k,n} = \frac{1}{\alpha_n \eta_n} \frac{C_{k,n}^{-*}}{X_n^*}.$$

Finally, we rewrite the normalized equation as

$$\dot{x}_n = \alpha_n \Big( \beta_n b_n + \widehat{\beta}_n q_n + \overline{\beta}_n u_n + \widetilde{\beta}_n \sum_k \sigma_{k,n} c_{k,n}^+ - \eta_n \sum_k \lambda_{k,n} c_{k,n}^- - \overline{\eta}_n l_n \Big).$$

**Governance Institution Equations**

We write the normalized function

$$i_m^+(y_m, K_{1,m}^+ x_1, \dots, K_{N,m}^+ x_N) := \frac{I_m^+(y_m Y^*, K_{1,m}^+ x_1 X_1^*, \dots, K_{N,m}^+ x_N X_N^*)}{I_m^{+*}},$$

and likewise for $i_m^-$.

This allows us to rewrite the equation for $\dot{Y}_m$ in terms of the normalized variables and functions as

$$\dot{y}_m = \frac{I_m^{+*}}{Y^*} i_m^+ - \frac{I_m^{-*}}{Y^*},$$

which leads us to define the scale parameter

$$\mu_m := \frac{I_m^{+*}}{Y^*} = \frac{I_m^{-*}}{Y^*}.$$

Finally, we rewrite the normalized equation as

$$\dot{y}_m = \mu_m \left( i_m^+ - i_m^- \right).$$

**Jacobian and Exponent Parameters**

We find the relevant *exponent parameters* by looking at the corresponding entries of the Jacobian.

**From the Resource Equation**

$$\frac{\partial \dot{r}}{\partial r} = \phi \left( \frac{\partial s}{\partial r} - \sum_n \psi_n \frac{\partial e_n}{\partial r} \right)$$

$$\frac{\partial \dot{r}}{\partial x_i} = -\phi \sum_n \psi_n \sum_m \frac{\partial e_n}{\partial g_{m,n}} \cdot \frac{\partial g_{m,n}}{\partial (F_{i,m,n} x_i)} \cdot F_{i,m,n}$$

$$\frac{\partial \dot{r}}{\partial y_m} = -\phi \sum_n \psi_n \frac{\partial e_n}{\partial g_{m,n}} \cdot \frac{\partial g_{m,n}}{\partial y_m}$$

**From the Resource User and Non-Resource User Actor Equations**

$$\frac{\partial \dot{x}_n}{\partial r} = \alpha_n \left( \beta_n \frac{\partial b_n}{\partial e_n} \cdot \frac{\partial e_n}{\partial r} + \widehat{\beta}_n \frac{\partial q_n}{\partial a_n} \cdot \frac{\partial a_n}{\partial r} \right)$$

For $i \neq n$:

$$\frac{\partial \dot{x}_n}{\partial x_i} = \alpha_n \left( \beta_n \frac{\partial b_n}{\partial e_n} \cdot \sum_m \frac{\partial e_n}{\partial g_{m,n}} \cdot \frac{\partial g_{m,n}}{\partial (F_{i,m,n} x_i)} \cdot F_{i,m,n} \right.$$
$$+ \widehat{\beta}_n \frac{\partial q_n}{\partial a_n} \cdot \sum_m \frac{\partial a_n}{\partial p_{m,n}} \cdot \frac{\partial p_{m,n}}{\partial (H_{i,m,n} x_i)} \cdot H_{i,m,n}$$
$$\left. + \widetilde{\beta}_n \sigma_{i,n} \frac{\partial c_{i,n}^+}{\partial (W_{i,n}^+ x_i)} W_{i,n}^+ - \eta_n \lambda_{i,n} \frac{\partial c_{i,n}^-}{\partial (W_{i,n}^- x_i)} W_{i,n}^- \right)$$

For $i = n$:

$$\frac{\partial \dot{x}_n}{\partial x_n} = \alpha_n \left( \beta_n \frac{\partial b_n}{\partial e_n} \cdot \sum_m \frac{\partial e_n}{\partial g_{m,n}} \cdot \frac{\partial g_{m,n}}{\partial (F_{n,m,n} x_n)} \cdot F_{n,m,n} \right.$$
$$+ \widehat{\beta}_n \frac{\partial q_n}{\partial a_n} \cdot \sum_m \frac{\partial a_n}{\partial p_{m,n}} \cdot \frac{\partial p_{m,n}}{\partial (H_{n,m,n} x_n)} \cdot H_{n,m,n}$$
$$\left. + \overline{\beta}_n \frac{\partial u_n}{\partial x_n} - \overline{\eta}_n \frac{\partial l_n}{\partial x_n} \right)$$

$$\frac{\partial \dot{x}_n}{\partial y_m} = \alpha_n \left( \beta_n \frac{\partial b_n}{\partial e_n} \cdot \frac{\partial e_n}{\partial g_{m,n}} \cdot \frac{\partial g_{m,n}}{\partial y_m} + \widehat{\beta}_n \frac{\partial q_n}{\partial a_n} \cdot \frac{\partial a_n}{\partial p_{m,n}} \cdot \frac{\partial p_{m,n}}{\partial y_m} \right)$$

**From the Governance Institution Equations**

$$\frac{\partial \dot{y}_m}{\partial r} = 0$$

$$\frac{\partial \dot{y}_m}{\partial x_i} = \mu_m \left[ \frac{\partial i_m^+}{\partial (K_{i,m}^+ x_i)} K_{i,m}^+ - \frac{\partial i_m^-}{\partial (K_{i,m}^- x_i)} K_{i,m}^- \right]$$

$$\frac{\partial \dot{y}_m}{\partial y_m} = \mu_m \left[ \frac{\partial i_m^+}{\partial y_m} - \frac{\partial i_m^-}{\partial y_m} \right]$$

For $j' \neq m$:

$$\frac{\partial \dot{y}_m}{\partial y_{j'}} = 0$$

**Derivation of Objective Function Gradient**

At equilibrium, the equation

$$\frac{d}{d\mathfrak{p}} \begin{pmatrix} r^* \\ x_n^* \\ y_m^* \end{pmatrix} = -J^{-1} \begin{pmatrix} \frac{\partial \dot{r}}{\partial \mathfrak{p}} \\ \frac{\partial \dot{x}_n}{\partial \mathfrak{p}} \\ \frac{\partial \dot{y}_m}{\partial \mathfrak{p}} \end{pmatrix}$$

describes how the steady state changes with respect to a strategy parameter $\mathfrak{p}$. The following sections show the calculation of the right-hand side of this equation for each of the strategy parameters.

**Calculation of Right-Hand Side**

**Calculations for** $F_{k,m,n}$

$$\frac{\partial \dot{r}}{\partial F_{k,m,n}} = -\phi \psi_n \frac{\partial e_n}{\partial g_{m,n}} \cdot \frac{\partial g_{m,n}}{\partial (F_{k,m,n} x_k)}$$

$$\frac{\partial \dot{x}_i}{\partial F_{k,m,n}} = \begin{cases} \alpha_n \beta_n \frac{\partial b_n}{\partial e_n} \cdot \frac{\partial e_n}{\partial g_{m,n}} \cdot \frac{\partial g_{m,n}}{\partial (F_{k,m,n} x_k)} & \text{if } i = n \\ 0 & \text{if } i \neq n \end{cases}$$

$$\frac{\partial \dot{y}_j}{\partial F_{k,m,n}} = 0$$

**Calculations for** $H_{k,m,n}$

$$\frac{\partial \dot{r}}{\partial H_{k,m,n}} = 0$$

$$\frac{\partial \dot{x}_i}{\partial H_{k,m,n}} = \begin{cases} \alpha_n \widehat{\beta}_n \frac{\partial q_n}{\partial a_n} \cdot \frac{\partial a_n}{\partial p_{m,n}} \cdot \frac{\partial p_{m,n}}{\partial (H_{k,m,n} x_k)} & \textbf{if } i = n \\ 0 & \textbf{if } i \neq n \end{cases}$$

$$\frac{\partial \dot{y}_j}{\partial H_{k,m,n}} = 0$$

**Calculations for** $W_{k,n}^{+}$ **and** $W_{k,n}^{-}$

$$\frac{\partial \dot{r}}{\partial W_{k,n}^{+}} = 0$$

$$\frac{\partial \dot{x}_i}{\partial W_{k,n}^{+}} = \begin{cases} \alpha_n \widetilde{\beta}_n \sigma_{k,n} \frac{\partial c_{k,n}^{+}}{\partial (W_{k,n}^{+} x_k)} & \text{if } i = n \\ 0 & \text{if } i \neq n \end{cases}$$

$$\frac{\partial \dot{y}_j}{\partial W_{k,n}^{+}} = 0$$

$$\frac{\partial \dot{r}}{\partial W_{k,n}^{-}} = 0$$

$$\frac{\partial \dot{x}_i}{\partial W_{k,n}^{-}} = \begin{cases} -\alpha_n \eta_n \lambda_{k,n} \frac{\partial c_{k,n}^{-}}{\partial (W_{k,n}^{-} x_k)} & \text{if } i = n \\ 0 & \text{if } i \neq n \end{cases}$$

$$\frac{\partial \dot{y}_j}{\partial W_{k,n}^{-}} = 0$$

**Calculations for $K_{k,m}^+$ and $K_{k,m}^-$**

$$\frac{\partial \dot{r}}{\partial K_{k,m}^+} = 0$$

$$\frac{\partial \dot{x}_i}{\partial K_{k,m}^+} = 0$$

$$\frac{\partial \dot{y}_j}{\partial K_{k,m}^+} = \begin{cases} \mu_m \dfrac{\partial i_m^+}{\partial (K_{k,m}^+ x_k)} & \text{if } j = m \\ 0 & \text{if } j \neq m \end{cases}$$

$$\frac{\partial \dot{r}}{\partial K_{k,m}^-} = 0$$

$$\frac{\partial \dot{x}_i}{\partial K_{k,m}^-} = 0$$

$$\frac{\partial \dot{y}_j}{\partial K_{k,m}^-} = \begin{cases} -\mu_m \dfrac{\partial i_m^-}{\partial (K_{k,m}^- x_k)} & \text{if } j = m \\ 0 & \text{if } j \neq m \end{cases}$$

**Calculating how objective functions change with each parameter**

**Extraction**

We have

$$\frac{de_n}{dF_{l,j,n}} = \frac{\partial e_n}{\partial r} \frac{\partial r^*}{\partial F_{l,j,n}} + \sum_m \left( \frac{\partial e_n}{\partial g_{m,n}} \frac{\partial g_{m,n}}{\partial y_m} \frac{\partial y_m^*}{\partial F_{l,j,n}} + \sum_k \frac{\partial e_n}{\partial g_{m,n}} \frac{\partial g_{m,n}}{\partial (F_{k,m,n} x_k)} \frac{\partial x_k^*}{\partial F_{l,j,n}} \cdot F_{k,m,n} \right)$$
$$+ \frac{\partial e_n}{\partial g_{j,n}} \frac{\partial g_{j,n}}{\partial (F_{l,j,n} x_l)}.$$

For any other effort allocation parameter $\mathfrak{p}$, including $\mathfrak{p} = F_{l,j,i}$ when $i \neq n$, we can use the general formula

$$\frac{de_n}{d\mathfrak{p}} = \frac{\partial e_n}{\partial r} \frac{\partial r^*}{\partial \mathfrak{p}} + \sum_m \left( \frac{\partial e_n}{\partial g_{m,n}} \frac{\partial g_{m,n}}{\partial y_m} \frac{\partial y_m^*}{\partial \mathfrak{p}} + \sum_k \frac{\partial e_n}{\partial g_{m,n}} \frac{\partial g_{m,n}}{\partial (F_{k,m,n} x_k)} \frac{\partial x_k^*}{\partial \mathfrak{p}} \cdot F_{k,m,n} \right).$$

**Accessz**

We have

$$\frac{da_n}{dH_{l,j,n}} = \frac{\partial a_n}{\partial r} \frac{\partial r^*}{\partial H_{l,j,n}} + \sum_m \left( \frac{\partial a_n}{\partial p_{m,n}} \frac{\partial p_{m,n}}{\partial y_m} \frac{\partial y_m^*}{\partial H_{l,j,n}} + \sum_k \frac{\partial a_n}{\partial p_{m,n}} \frac{\partial p_{m,n}}{\partial (H_{k,m,n} x_k)} \frac{\partial x_k^*}{\partial H_{l,j,n}} \cdot H_{k,m,n} \right)$$
$$+ \frac{\partial a_n}{\partial p_{j,n}} \frac{\partial p_{j,n}}{\partial (H_{l,j,n} x_l)}.$$

For any other effort allocation parameter $\mathfrak{p}$, we can use the general formula

$$\frac{da_n}{d\mathfrak{p}} = \frac{\partial a_n}{\partial r} \frac{\partial r^*}{\partial \mathfrak{p}} + \sum_m \left( \frac{\partial a_n}{\partial p_{m,n}} \frac{\partial p_{m,n}}{\partial y_m} \frac{\partial y_m^*}{\partial \mathfrak{p}} + \sum_k \frac{\partial a_n}{\partial p_{m,n}} \frac{\partial p_{m,n}}{\partial (H_{k,m,n} x_k)} \frac{\partial x_k^*}{\partial \mathfrak{p}} \cdot H_{k,m,n} \right).$$

**Parameter Values and Ranges**

Parameters are derived from the Generalized Modeling approach described above.

| Parameter | Interpretation | Range | Value |
|:---:|:---:|:---:|:---:|
| **Scale Parameters** | | | |
| $\phi$ | Rate of turnover in the resource, or inverse of characteristic time scale of resource | 0 to 1 | |
| $\psi_n$ | Share of extraction of resource by user $n$ | 0 to 1, $\sum_n \psi_n = 1$ | |
| $\alpha_n$ | Rate of turnover in the capacity of user $n$ | 0 to 1 | |
| $\beta_n$ | Share of actor $n$ capacity gain in response to resource extraction | $\beta_n + \widehat{\beta}_n + \widetilde{\beta}_n + \overline{\beta}_n = 1$ | |
| $\widehat{\beta}_n$ | Share of actor $n$ capacity gain in response to resource access conditions | $\beta_n + \widehat{\beta}_n + \widetilde{\beta}_n + \overline{\beta}_n = 1$ | |
| $\widetilde{\beta}_n$ | Share of actor $n$ capacity gain from collaborations | $\beta_n + \widehat{\beta}_n + \widetilde{\beta}_n + \overline{\beta}_n = 1$ | |
| $\overline{\beta}_n$ | Share of actor $n$'s capacity gain from "natural" gain (non-resource users only) | $\beta_n + \widehat{\beta}_n + \widetilde{\beta}_n + \overline{\beta}_n = 1$ | |
| $\sigma_{k,n}$ | Share of actor $n$'s collaboration gain from collaborating with actor $k$ | 0 to 1 | |
| $\eta_n$ | Share of actor $n$'s loss in capacity due to direct undermining by other actors | $1 - \overline{\eta}_n$ | |
| $\lambda_{k,n}$ | Share of actor $n$'s loss from being undermined by other actors attributed to actor $k$ | 0 to 1 | |
| $\overline{\eta}_n$ | Share of actor $n$'s loss in capacity due to "natural" decay | $1 - \eta_n$ | |
| $\mu_m$ | Rate of turnover in decision center $m$'s capacity | 0 to 1 | |
| **Exponent Parameters** | | | |
| $\dfrac{\partial s}{\partial r}$ | Sensitivity of resource regeneration to resource state | $-1$ to 1 | $-0.5$ |
| $\dfrac{\partial e_n}{\partial r}$ | Sensitivity of extraction by user $n$ to resource state | 1 to 2 | 1.5 |
| $\dfrac{\partial e_n}{\partial g_{m,n}}$ | Sensitivity of extraction by user $n$ to intervention by decision center $m$ (effectiveness of intervention) | $-1$ to 1 | - |
| $\dfrac{\partial g_{m,n}}{\partial (F_{i,m,n} x_i)}$ | Sensitivity of intervention in user $n$'s extraction by decision center $m$ to actions by actor $i$ (effectiveness of actors' support/resistance) | 0 to 2 | 1 |
| $\dfrac{\partial g_{m,n}}{\partial y_m}$ | Sensitivity of extraction intervention by decision center $m$ to their own capacity | 0 to 2 | 1 |
| $\dfrac{\partial p_{m,n}}{\partial y_m}$ | Sensitivity of resource access intervention by decision center $m$ to their own capacity | 0 to 2 | 1 |
| $\dfrac{\partial b_n}{\partial e_n}$ | Sensitivity of user $n$'s gain in capacity based on extraction to the amount of extraction | $-1$ to 1 | 0.5 |
| $\dfrac{\partial a_n}{\partial r}$ | Sensitivity of access by user $n$ to resource state | 0 to 2 | 1 |
| $\dfrac{\partial q_n}{\partial a_n}$ | Sensitivity of user $n$'s gain in capacity based on resource access to the level of resource access | $-1$ to 1 | 0.5 |
| $\dfrac{\partial a_n}{\partial p_{m,n}}$ | Effectiveness of intervention $p$ by decision center $m$ in changing access for resource user $n$ | $-1$ to 1 | - |

| | | | |
|---|---|---|---|
| $\dfrac{\partial p_{m,n}}{\partial (H_{i,m,n} x_i)}$ | Sensitivity of intervention by decision center $m$ to actions by actor $i$ (effectiveness of actors' support/resistance) | 0 to 2 | 1 |
| $\dfrac{\partial c_{i,n}^{+}}{\partial (W_{i,n}^{+} x_i)}$ | Sensitivity of actor $n$'s gain from collaboration to actor $i$'s collaboration efforts | 0 to 2 | 1 |
| $\dfrac{\partial c_{i,n}^{-}}{\partial (W_{i,n}^{-} x_i)}$ | Sensitivity of actor $n$'s loss in capacity to other actor $i$'s efforts to undermine them | 0 to 2 | 1 |
| $\dfrac{\partial l_n}{\partial x_n}$ | Sensitivity of actor $n$'s "natural" decay in capacity $l$ to their own capacity | 0.5 to 1 | 1 |
| $\dfrac{\partial u_n}{\partial x_n}$ | Sensitivity of non-resource user actor $n$'s self-growth in capacity to their own capacity | 0 to 1 | 0.5 |
| $\dfrac{\partial i_m^{+}}{\partial (K_{i,m}^{+} x_i)}$ | Sensitivity of decision center $m$'s gain in capacity to actor $i$'s actions; likewise for $\frac{\partial i_m^{-}}{\partial (K_{i,m}^{-} x_i)}$ | 0 to 2 | 1 |
| $\dfrac{\partial i_m^{+}}{\partial y_m}$ | Sensitivity of decision center $m$'s gain in capacity to their own capacity | 0 to 1 | 0.5 |
| $\dfrac{\partial i_m^{-}}{\partial y_m}$ | Sensitivity of decision center $m$'s loss in capacity to their own capacity | 0 to 1 | 1 |

**Supplementary Figures**

[Figure]

Figure S1: Correlation results including all forms of strategy parameters and all significant parameters. The inclusion of the different forms of strategy parameters allows for concluding that stability depends on the magnitude of effort allocated to the strategies rather than the sign or direction of the effort.

[Figure]

Figure S2: Effect of system size (number of actors and decision centers) and connectance on stability. The connectance shown is the total connectance, which is computed after the experiment rather than set beforehand due to the dependence of the connectance on actors' computed strategies. As a result, there is no data for some combinations of connectance and size.

[Figure]

Figure S3: Effect of different types of resource users (extractors, accessors, and combined extractors and accessors) on stability. The color represents the proportion of stable systems for a given system composition. The total system size is 10, with 8 resource users and 2 decision centers. The proportion of extractors as compared to accessors or combined extractors and accessors has no effect on stability.